# Effect of Feeding *Saccharomyces cerevisiae boulardii* CNCM I-1079 to Sows and Piglets on Piglets’ Immune Response after Vaccination against *Actinobacillus pleuropneumoniae*

**DOI:** 10.3390/ani12192513

**Published:** 2022-09-21

**Authors:** Fernando Bravo de Laguna, Carolina Cabrera, Ana Belén González, Clara de Pascual, Francisco José Pallarés, Eric Chevaux, Mathieu Castex, David Saornil, Pierre Lebreton, Guillermo Ramis

**Affiliations:** 1Lallemand SAS, 31702 Blagnac, France; 2Department of Animal Production, Faculty of Veterinary Medicine, University of Murcia, 30100 Murcia, Spain; 3Department of Anatomy and Comparative Pathology, Faculty of Veterinary Medicine, University of Córdoba, 14071 Córdoba, Spain

**Keywords:** *Saccharomyces cerevisiae boulardii*, gut-lung axis, maternal imprinting, cytokines, weanling piglets

## Abstract

**Simple Summary:**

The production of high-quality pigs starts with sow nutrition. In this study, we tested the effect of feeding the live yeast *Saccharomyces cerevisiae boulardii* CNCM I-1079 to sows and their piglets on litter performance and some immunity parameters of piglets vaccinated against *Actinobacillus pleuropneumoniae*. Our results indicate that feeding lactating sows with the live yeast improves litter growth and weaning weight. For that reason, the pigs are likely better prepared for the challenges they are going to encounter after weaning. The piglets from supplemented sows show higher mucosal protection and less inflammation in the lung after the vaccine, explained by the gut–lung axis, which is defined as the cross-talk between the gut microbiota and the lung.

**Abstract:**

The aim of this study was to assess the effect that feeding *Saccharomyces cerevisiae boulardii* CNCM I-1079 (LSB) to lactating sows and their progeny has on inflammatory response and mucosal immunity after vaccination against *Actinobacillus pleuropneumoniae.* Sixty-seven Danbred sows were allotted into two treatments when they entered the farrowing room seven days before the expected farrowing date: control (CON: lactation diet) and LSB (CON + 12 × 10^9^ colony forming units (CFU)/d until weaning). At weaning, piglets were equally allotted into two experimental diets according to sow diet: control (CON: 2-phase post-weaning diets) and LSB (CON + 2 × 10^9^ CFU/kg and 1 × 10^9^ CFU/kg in phases 1 and 2, respectively). The piglets were vaccinated at days 26 and 49 post-weaning. Growth performance and number of IgA producing cells and cytokine’s gene expression in lung, lymph node, and intestine samples at day 70 post-weaning were assessed and analyzed in SPSS Statistics 26: performance with a general linear model with sex, room, sow diet, piglet diet, and their interactions as main effects, and immunity with a Kruskal–Wallis test for k unrelated samples. Piglets from LSB-fed sows displayed a higher average daily gain (ADG; *p* < 0.01) and a heavier body weight (BW; *p* < 0.05) during lactation, tended (*p* < 0.1) to be heavier at day 49, and to have a higher ADG between days 26 and 49; had fewer number of IgA producing cells in the lymph node (*p* < 0.05); and all the cytokines studied were significantly under-regulated (*p* < 0.05) in the lung. It is concluded that feeding *Saccharomyces cerevisiae boulardii* CNCM I-1079 to sows improved piglet performance during lactation and showed a clear reduction in the inflammatory status of the lungs after vaccination against *A. pleuropneumoniae*, suggesting that there was a maternal imprinting effect on mucosal protection and a cross-talk between the gut microbiota and the lung.

## 1. Introduction

The production of high-quality pigs starts with sow nutrition. Gestation diets affect reproductive performance and birth weight [1], whereas lactation diets affect litter performance and weaning weight [2,3]. Especially with modern hyperprolific genetics producing piglets with a significant reduction in birth weight [4], it becomes very important to implement strategies to ensure a good lactation performance to obtain robust piglets at weaning are ready to face all the subsequent challenges, understanding robustness not only as weight-related, but also to the level of development of the immune system.

Feeding probiotics to lactating sows has positive effects on litter performance. This effect can come from an increase in sow feed intake [5], a decrease in farrowing duration which makes piglets more vital [5], or by affecting colostrum yield or composition [6]. Other yeast derivatives such as hydrolyzed yeast fed during pregnancy have also shown effects on colostrum yield and fecal microbiota of both sows and neonate piglets [7].

Regarding immunity, feeding probiotics to lactating sows affects piglet’s immunity acquisition during lactation. For example, colostrum of sows fed *Saccharomyces cerevisiae boulardii* CNCM I-1079 increases its concentration in IgG and IgA, improving the transfer of passive immunity to the neonates [8,9], thus improving their ability to respond to challenges posed by pathogens. Cytokines are inflammatory biomarkers critical for humoral and cell mediated immunity, and their expression may vary with factors like weaning, vaccines, or probiotic supplementation. Stimulation of the pig’s intestinal epithelial cells up-regulates cytokine transcripts, and probiotics help in the cytokine production by T-cells, when the expression of pro-inflammatory cytokines such as TNF-α is down-regulated [10]. Additionally, mucosal immunity is based on IgA production, as it is the main immunoglobulin of the mucus layer. Therefore, an intervention aiming at increasing mucosal protection should be able to show up a higher number of IgA production cells. 

However, research about the effects of lactating sow’s supplementation with probiotics in general, and *S. cerevisiae boulardii* CNCM I-1079 in particular, on piglets’ immune status several weeks after weaning is scarce. This concept of maternal imprinting opens a new field of research since piglets can be cared from conception instead of from weaning. Furthermore, the challenge of modulating common respiratory diseases through products that are orally administered to the sow deserves attention.

We hypothesize that feeding lactating sows with *S. cerevisiae boulardii* CNCM I-1079 affects the performance and immune response of piglets after weaning due to the maternal imprinting concept, and that there is an effect in the respiratory tract due to the gut–lung axis. The aim of this study was to assess the effect that feeding *Saccharomyces cerevisiae boulardii* CNCM I-1079 to lactating sows and their piglets after weaning has on the inflammatory response and mucosal immunity of piglets after vaccination against *Actinobacillus pleuropneumoniae.*

## 2. Materials and Methods

The experiment took place in a commercial farm in the region of Murcia (Spain), under the supervision of the Faculty of Veterinary Science of the University of Murcia. Before starting the trial, random blood samples were taken from the sows to check for the absence of circulating antibodies and *A. pleuropneumoniae.* The results were negative.

### 2.1. Animals, Diets and Measurements

In a 2 × 2 factorial design, 67 Danbred sows between parities 2–4 (2.88 ± 0.87; mean ± SD) from 2 subsequent batches were allocated into 2 treatments according to their parity number and backfat thickness (BFT; Renco Lean-Meater, Renco Corporation, Minneapolis, MN, USA) when they entered the farrowing room 7 days before the expected farrowing date: control (CON: commercial lactation diet), and LSB (CON + 2 × 10^9^ colony forming units (CFU)/kg feed of the live yeast *Saccharomyces cerevisiae boulardii* CNCM I-1079 (Levucell^®^ SB 2; Lallemand SAS, Blagnac, France) during the morning meal until weaning. Mean parity number was 2.91 ± 0.86 and 2.86 ± 0.88 for CON and LSB sows, respectively). Five slatted-floor farrowing rooms of 16 crates each were used in the trial. Sows were fed twice a day, and daily feed intake (ADFI) per sow was assessed during lactation, by weighting the remaining feed 2 h after each meal. If the sows had eaten all the portion, the subsequent meal was increased by 200 g. Otherwise, the feed allowance was maintained. Litters were homogenized by size within treatment 24 h after birth, when the piglets were individually weighted as well as at weaning. Average daily gain (ADG) was calculated as the difference between weaning weight and initial weight, divided by the number of days in lactation. The number of weaned piglets was recorded. At weaning, piglets from batch 1 were moved to the post-weaning facilities and separated in 2 groups according to sow diet during lactation. They were randomly distributed by BW in 4 rooms of 10 pens each, in mixed-sexes pens of 18–20, so that average initial BW of the piglets within a pen was as much similar as possible, filling 32 pens in total (8 pens/room, leaving 2 pens free for sick or out of trial piglets). Pens were equally allocated into 2 experimental diets according to initial BW: control (CON: commercial post-weaning diets) and LSB (CON + 2 × 10^9^ CFU/kg feed and 1 × 10^9^ CFU/kg feed in Prestarter and Starter, respectively, of the live yeast *S. cerevisiae boulardii* CNCM I-1079 (Levucell^®^ SB 10ME Titan, Lallemand SAS, Blagnac, France). Piglets followed a 2-phase feeding program: Prestarter (from weaning until day 18), and Starter (between days 19 and 70 post-weaning). All the experimental diets were formulated to meet or exceed FEDNA requirements [11] (Table 1 and Table 2). 

Prestarter diets were medicated with 1500 ppm ZnO. All the piglets were vaccinated against *A. pleuropneumoniae* (APP; Coglapix, Ceva Santé Animale, Libourne, France) on days 26 and 49 post-weaning, as indicated by manufacturer. Fifteen piglets per group were selected according to the “average piglet” criteria on the first vaccination day, and they were individually weighed at each vaccination day. At day 42 post-weaning, the piglets were moved to the fattening facility, where they were kept allocated by treatment and fed the Starter diets for 28 extra days, when the trial finished at day 70 post-weaning (i.e., 3 weeks after the vaccination booster).

At the end of the experiment, the selected pigs were moved to the necropsy room at the Faculty of Veterinary Science of the University of Murcia, where they were individually weighed and humanely euthanized using an overdose of thiobarbital IV (50 mg/kg BW, Thiobarbital Braun Medical S.A., Barcelona, Spain). Animals were sampled for lung, mediastinal lymph node, and jejunum. ADG was calculated in the period between vaccinations, and from the first vaccination until the end of the trial. Tissue samples were assessed for the presence of IgA producing cells, and for cytokines’ gene expression.

### 2.2. IgA Producing Cells Presence Assessment

The tissue samples were fixed in Bouin’s solution. The samples were processed routinely for paraffin-embedded tissue production into the first week after sampling. A series of 4 μm slides were stained and immunohistochemistry was applied following the Avidin Biotin Complex (ABC) technique. Goat anti-Pig IgA Antibody Affinity Purified (Bethyl Laboratories Inc., Montgomery, TX, USA) was used as primary antibody incubated overnight, and a Polyclonal Goat Anti-Rabbit Immunoglobulins/Biotinylated (Dako, Glostrup, Denmark) as secondary. The Vectastain Elite ABC kit (Vector Laboratories Inc., Newark, CA, USA) was applied for 1 h at 20 °C, and positive labeling was detected using 3,3′-diaminobenzidine tetrahydrochloride (Dako, Glostrup, Denmark). The tissue sections were counterstained with Mayer’s hematoxylin, dehydrated, and mounted. Once identified, IgA producing cells were counted by images software analysis under a Zeiss Axioskop 40 light microscope (Carl Zeiss AG, Oberkochen, Germany; Figure 1). The count was the average of the cells counted in 10 non-overlapped fields of 10.000 μm^2^.

### 2.3. Gene Expression for Cytokines

A total of 11 cytokines: IFN-α, IFN-γ, TNF-α, IL-12p35, IL-12p40, IL-10, TGF-β, IL-8, IL-1α, IL-1β, and IL-6 were analyzed, using primers previously described by other authors (Table 3). The tissue samples were preserved in RNAlater (1 mL/20 mg) and frozen at −27 °C up to analysis after 24 h of refrigeration. The mRNA was isolated using a commercial kit (RNeasy Micro kit, Qiagen, Germantown, TN, USA) and cDNA was synthetized to perform q-PCR using SYBR-green chemistry. The relative quantification was conducted using β-actin as housekeeping gene. Data were expressed relatively as fold change and log-2 transformed, normalizing each expression to the lowest value obtained for a specific cytokine in a tissue among all the samples, which was assigned a value 1.

### 2.4. Statistical Analysis

Data were analyzed in SPSS Statistics 26.0 (IBM). All the variables were checked for normality according to the test of Kolmogorov–Smirnov. Piglet’s performance and sows’ ADFI were analyzed by analysis of variance using a general linear model with parity number, treatment, and their interactions as main effects. BW at birth was used as a covariate to analyze piglet’s performance. The experimental unit was the sow for ADFI and the litter for piglet’s performance. Post-weaning, the experimental unit was the piglet. Piglets’ body weight and growth were submitted to analysis of variance with sex, room, sow diet, piglet diet, and their interactions as main effects. BW at day 26 post-weaning was used as a covariate. The comparison among groups of IgA producing cells and gene expression for cytokines were analyzed with Kruskal–Wallis test for k unrelated samples. The correlations between IgA producing cells in different tissues was calculated by Spearman’s correlation analysis. Differences with a *p*-Value < 0.05 were considered as significant, and *p*-Values between 0.05 and 0.1 were considered a trend.

## 3. Results

### 3.1. Sows’ Performance

There were significant differences in ADG *(p* < 0.01) and final BW (*p* < 0.05) of the piglets, with the piglets from LSB-fed sows the ones showing faster growth and a heavier BW (Table 4).

### 3.2. Piglets’ Performance

There was no effect of sex and no interaction between sex and treatment, therefore, those effects were removed from the model. Since there was no interaction between sow diet and piglet diet, both effects were analyzed separately (Table 5). Piglets from LSB-fed sows tended (*p* < 0.1) to be heavier at day 49 post-weaning and grew faster between days 26 and 49 post-weaning. Furthermore, LSB-supplemented piglets displayed a heavier BW at day 49 post-weaning (*p* < 0.05) and a faster ADG between days 26 and 49 post-weaning (*p* < 0.05). 

### 3.3. IgA Producing Cells

We observed a significant difference (*p* < 0.05; Table 6) in the number of IgA producing cells in the lymph node, with piglets from LSB-fed sows displaying a lower number. Consistently, LSB-fed piglets tended (*p* < 0.1; Table 6) to have a lower number of IgA producing cells in the lymph node compared to CON piglets. 

There was a significant and positive correlation between the quantification in the intestine and in the lung (*p* < 0.001; R = 0.430; Figure 2) and between the lung and the lymph node (*p* < 0.01; R = 0.372; Figure 3).

### 3.4. Cytokines Gene Expression in Tissues

Piglets from LSB-fed sows showed a trend toward a higher expression of IL-1α (*p* < 0.1) in the lymph node compared to piglets from CON sows. In the lung, all the cytokines were significantly underregulated (*p* < 0.05; Table 7) in the piglets from LSB-fed sows. No difference was found in the intestine.

## 4. Discussion

The present study aimed at assessing the effects live yeast *S. cerevisiae boulardii* CNCM I-1079 (LSB) supplementation to lactating sows may have on their offsprings that were subsequently receiving the same live yeast during the post-weaning phase. Growth performance, IgA producing cells, and key tissue cytokines’ gene expressions were used to appraise this effect while piglets were challenged with APP vaccination. We found a significant difference in weaning weight and lactation growth of suckling piglets from LSB-fed sows. This result is in line with previous observations in lactating sows [6,19,20], even though the duration of the supplementation in the current study was shorter than reported in the literature.

The effects we observe in the post-weaning phase are likely a direct consequence of the metabolic status and robustness that piglets from LSB-fed sows express at weaning. We measured individual BW and growth between the first vaccination and slaughter. We found differences in BW at the second vaccination and in growth between the first and the second vaccination, where the LSB-supplemented piglets displayed a heavier BW and a faster growth compared to the CON-fed piglets. Furthermore, looking at the effects of sow diet on performance, we could observe that piglets from LSB-fed sows were heavier at the second vaccination and grew faster between both vaccinations than their CON counterparts. A preliminary explanation could come from the stress caused by transport and environmental changes in the period between vaccinations when the animals were moved to the fattening facility. The CON-fed piglets may be less prepared to cope with those changes and manage the stress, which directly affected their performance. That is in line with observations in LSB-supplemented sows around the stressful event of farrowing, with reduction of the farrowing duration [5], a clear indication of lower stress. After the second vaccination (one week after transportation to the fattening facility), the performance data suggest that there was an adaptation of the CON-fed piglets to the new conditions, and the differences became non-significant.

A second explanation could lie in the inflammatory response. The piglets from LSB-fed sows displayed both a lower mucosal immune and anti-inflammatory responses; and the expression of the immune biomarkers is not necessarily correlated with an improvement in performance, since their production may involve an increase in energy requirements for the animals [21]. Therefore, the lower cytokine expression and number of IgA producing cells in the piglets from LSB-fed sows could partially explain the improved performance between both vaccinations since more energy would be available for growth. Indeed, although the gene expressions in our study were recorded only at day 70 post-weaning and we do not have a track of the inflammatory response between days 26 and 49 post-weaning, it could be that the increased cytokines’ expression displayed by the pigs from CON sows was already evident in previous weeks. If we assume that hypothesis, the lack of difference in performance between days 49 and 70 post-weaning could be explained by the lower immune activation of the piglets from LSB-fed sows, which coinciding with the transport to the new facility might be less ready to face new challenges like opportunistic pathogens, new environment, new pen-mates, and the vaccine administered. De Groot et al. [22] reported that inflammatory response after weaning is transient, reverting later to a pre-weaning situation. Considering the period after the second vaccination as a stress comparable to weaning, all the pigs may have been affected to the same extent causing no differences in performance between days 49 and 70 post-weaning. Anyway, since three weeks after the second vaccination the piglets from CON-fed sows displayed a higher expression in the lung than the piglets from LSB-fed sows, it is proposed that LSB supplementation to sows may have helped their piglets to speed up the process to recover the basal inflammatory status after the second vaccination, reverting earlier to the pre-vaccination situation, in line with the findings of de Groot et al. [22].

In most animals, the mucosal immune system is responsible for the active immune responses control [23]. The mucosal immunity is mainly based on IgA response [24,25]. Therefore, any vaccine or action focused on mucosal defense should be able to produce a remarkable IgA concentration, as is the case of the APP vaccine in our study. We found fewer IgA producing cells in the lymph node of both LSB-fed piglets and piglets from LSB-fed sows. One may think that the mucosal immunity of those piglets was lower; however, mucosal immunity develops in response to a challenge after the antigen presentation by the dendritic cells or the reaction to the release of several cytokines [24]. Therefore, we could hypothesize that the lower IgA producing cells found in those animals indicates that they were less challenged in terms of contact with an immune stimulating organism/agent, suggesting higher mucosal protection explained by the LSB supplementation to the sows and the own piglet’s intake. Furthermore, there was a positive correlation in the number of IgA producing cells between jejunum and lung, which could suggest that the lower contact with the immune stimulating organism started in the gut. The reduction in IgA in piglets supplemented with a probiotic was already reported [26].

The decreased cytokine expression found in our study in the piglets from LSB-fed sows could be attributed to the dietary treatments because we know from Czyżewska-Dors et al. [27] that an infection with APP in seven-week-old piglets increased lung concentrations of IL-1β, IL-6, IL-8, and IFN-γ. Therefore, and due to the importance of cytokines in immunological and inflammatory reactions [28], the reduced expression indicates a modulation of the inflammatory response. The production of IgA is controlled by both the T-cells, which also produce cytokines, and the cytokines directly released by the mucosa [22]. Thus, the lower level of cytokines observed in our study is in line with the lower number of IgA producing cells. Although the decrease in cytokines and IgA producing cells were found in different tissues (lung and lymph node, respectively), we can believe they are linked since we found a positive correlation in the number of IgA producing cells between lung and lymph node, which could be expected since the mediastinal lymph node is in the lung region. At this stage of the discussion, two concepts are proposed to explain the fact that an orally-administered live yeast affects immune variables in the respiratory tract, the reduction in the IgA producing cells in both LSB-fed piglets and piglets from LSB-fed sows, the correlation in the number of IgA producing cells between jejunum and lung, and the observation of a lower inflammation suggested by the cytokine expression in the lung of piglets from LSB-fed sows after a respiratory challenge such as the APP vaccine: (i) the connection between the gastrointestinal and respiratory tracts (gut–lung axis) [29] and (ii) the maternal imprinting concept. Considering that LSB supplemented to sows affects piglet microbiota even after weaning [30], it brings together both concepts to better understand our results.

Maternal imprinting is the set of events affecting the mother that will show up in the next generations. It is found in different characteristics of the offspring, i.e., performance, microbiota, or immunity. Social, environmental, or nutritional factors can play an important role on offspring’s development. If we focus on nutrition during lactation, it is widely accepted that the composition and quality of colostrum and milk is key for the suckling piglets to reach weaning with a better metabolic and health status, and immune development. Regarding immunity, the effects of sow nutrition on piglet immunity up to 70 days post-weaning as we measured in our study are not described. However, the concentration of some cytokines in sow serum is correlated with that of colostrum/milk, and at the same time with the concentration in piglet serum [31], suggesting that any intervention on the sows to increase their circulating cytokines level can affect the reaction of the piglets to an inflammatory process, not only in the short term but also in later stages, as we have observed. Colostrum plays an important role in the microbiota establishment of the suckling piglet with effects that the maternal diet may have on the progeny, as its quality may change with sow diet. Feeding hydrolyzed yeast during gestation increased the microbial diversity of suckling piglets one week after birth as well as the number of beneficial microbial populations [7]. Furthermore, LSB supplemented to sows affects gut microbiota of piglets after weaning [30], which could be due to the contact of the piglets with sow feces during lactation, or to changes in colostrum composition, since LSB affects colostrum nutritional [6] and immunological [8] composition.

The gut–lung axis is defined as the bi-directional cross-talk between gut and lung [29,32]. This communication is driven by the gut microbiota, through the changes that the microbes cause to the host’s metabolism. A dysbiosis of the gut microbiota may be the origin of health issues in both the gut and in distal organs [33,34]. More specifically, there are proven connections between gut microbiota and lung immunity [35]. Gut microbiota protects against respiratory infection in mice and humans, and its reduction impairs immune responses [29,36]. In swine, a co-infection with porcine reproductive and respiratory virus (PRRS) and porcine circovirus type 2 (PCV2) differently affected piglets in terms of viremia and growth performance [37]. Additionally, the fecal microbial diversity 70 days post-infection was decreased in the low performing piglets, which indicates a key role of gut microbiota in the clinical signs. Looking at the role of microbiota at the moment of the co-infection instead of 70 days post-infection, the microbial diversity was already increased prior to infection in the animals showing less lung lesions, reduced virus replication and faster average daily gain after the exposure to the virus [38], indicating that gut microbiota predisposes the animals to a specific response to the respiratory disease. In our study, although there were no microbiota measurements, if we assume that the microbiota of the piglets from LSB-fed sows in our study differed compared to the one of piglets from CON sows as previously reported [30], we could partially interpret our results as an effect of the gut microbiota to diminish the effects of the respiratory challenge of the vaccine in the lungs, showing differences in lung inflammation and mucosal protection of the piglet. There is also evidence of differences in gut microbiota among animals vaccinated against PRRS and co-infected with PCV2 [39]. Differences in microbiota led to different performances, suggesting an effect on vaccination efficacy. In our study, vaccination efficacy could not be measured and would require further research. However, there is evidence about the existence of maternal imprinting in the vaccination efficacy [40]. Feeding short chain fructo-oligosaccharides (scFOS) to either the sow at the end of gestation and during lactation or to the piglets during post-weaning, the humoral response after vaccination against influenza was increased, and piglets showed a higher level of specific IgA levels for influenza in both serum and feces.

Immunomodulatory feed additives such as probiotics are used to limit the possible failures of the immune response and reduce inflammation, as observed in our study. The idea behind the use of probiotics and prebiotics is to improve the function of the immune system, by either the competitive exclusion of the pathogen or by activating the response of any immunological component [23]. Bissonnette et al. [41] used lipopolysaccharide (LPS) challenged weanling piglets to assess the effect of feed additives (including prebiotics) on inflammatory response by measuring blood-circulating cytokines and gene expression in the ileum. Having shown that the challenge induced an inflammation, their results suggested a lower gene response in the piglets offered the feed supplements, in line with our findings. Early post-challenge, the release of pro-inflammatory cytokines helped in the defense against pathogens, although high levels cause tissue over-inflammation and even destruction. Therefore, the addition of LSB may have balanced both reactions by inhibiting the adhesion of pathogens and attenuating and overreacting inflammatory response [42]. Baker’s yeast *Saccharomyces cerevisiae* and its derivatives have been reported as immunomodulatory compounds, by affecting directly and indirectly the immune system and its biomarkers [21]. In line with our study, *Saccharomyces cerevisiae* inhibited pro-inflammatory gene expression.

## 5. Conclusions

It is concluded that feeding *S. cerevisiae boulardii* CNCM I-1079 to sows improved piglet performance during lactation and showed a clear effect in mitigating piglets’ inflammation response in the lungs after vaccination against *A. pleuropneumoniae*, suggesting that there was a maternal imprinting effect on mucosal protection. Because of the lower inflammation observed, animals displayed a faster growth derived from the lower energy costs required for the inflammatory reaction. 

We hypothesize that feeding the live yeast *S. cerevisiae boulardii* CNCM I-1079 to sows from one week before farrowing until weaning modulates the gut microbiota of piglets, reducing the concentration of pathogenic bacteria in the digestive tract, therefore reducing inflammation and consequently the secretion of cytokines in the lung. The specific metabolic mechanisms of the cross-talk between the gut microbiota and respiratory tract needs to be further elucidated.

## Figures and Tables

**Figure 1 animals-12-02513-f001:**
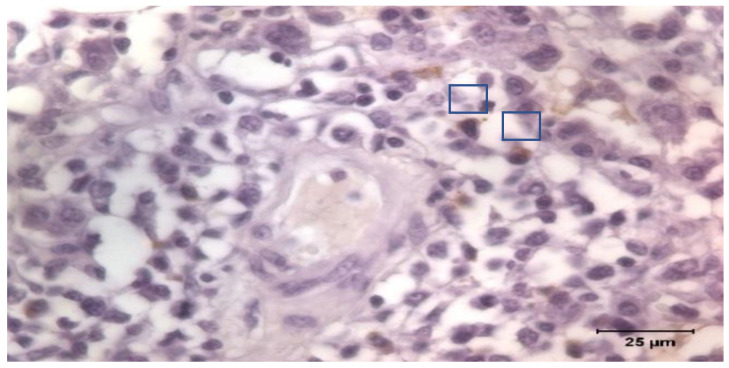
Immunostaining of IgA producing cells in lung. Detail of stained cells.

**Figure 2 animals-12-02513-f002:**
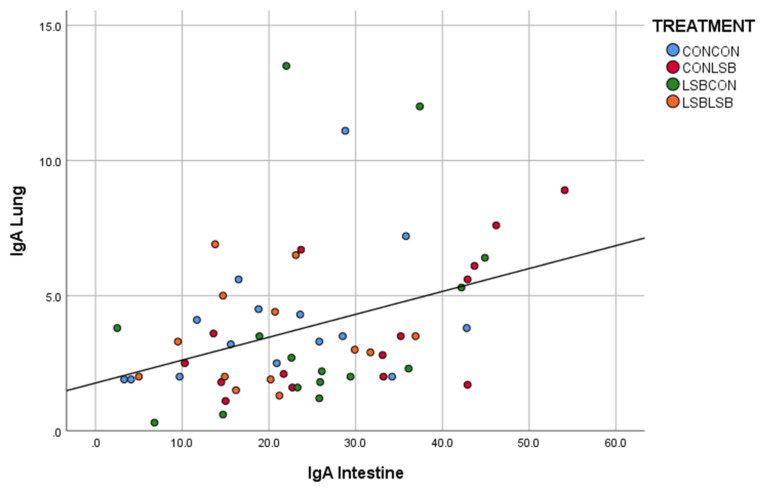
Correlation between the number of IgA producing cells in the intestine and in the lung.

**Figure 3 animals-12-02513-f003:**
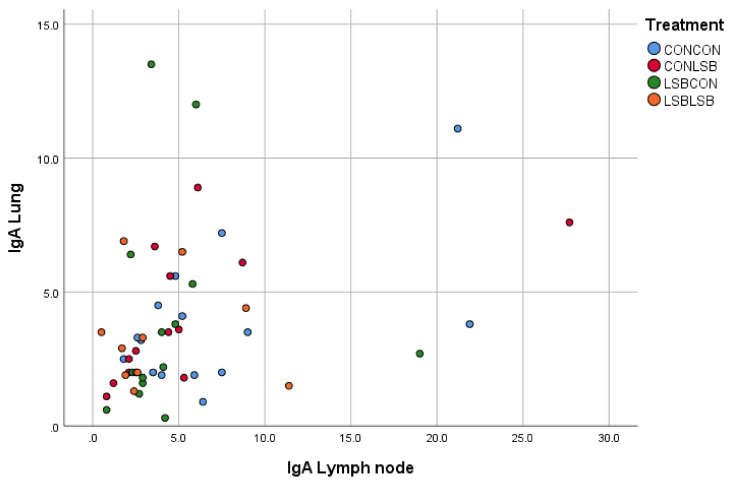
Correlation between the number of IgA producing cells in the lymph node and in the lung.

**Table 1 animals-12-02513-t001:** Composition of the control experimental diets (%, as fed).

Ingredient	Lactation	Prestarter	Starter
Lactose-enriched extruded wheat	-	12.00	-
Barley	33.76	15.50	20.45
Soybean meal 47	15.38	11.00	20.00
Corn	15.00	16.02	24.80
Wheat	15.00	16.92	24.70
Beet pulp	6.06	-	-
Sunflower meal 35	4.00	-	-
Sweet whey	-	6.67	-
Extruded wheat	-	2.50	-
Extruded corn	-	2.50	-
Fat 3/5	3.06	-	-
Wheat bran	3.00	-	-
Fish meal	-	5.00	-
Animal plasma	-	1.33	-
Full fat soybean	-	3.00	2.00
Sodium bicarbonate	0.53	-	-
Soy oil	-	2.07	3.06
Hydrolyzed pig protein	-	1.67	1.00
Feed preservative	0.20	-	-
Lignocellulose	-	1.33	-
Mineral premix	0.30	0.60	0.30
L-Lys	0.65	0.48	0.58
Salt	0.40	0.27	0.40
L-Thr	0.17	0.21	0.23
Calcium Carbonate	1.59	0.27	0.93
Monocalcium phosphate	0.60	0.20	0.67
Monobutyrin	-	-	0.13
DL-Met	0.11	0.20	0.18
Benzoic acid	-	-	0.25
Soy lecithin	-	0.10	0.10
L-Val	0.13	0.07	0.11
L-Trp	0.02	0.07	0.08
Choline chloride 78	0.03	-	-
Phytase 5000 P	0.02	-	-

**Table 2 animals-12-02513-t002:** Calculated nutritional composition of the control experimental diets (%, as fed).

Nutrient	Lactation	Prestarter	Starter
Dry matter	88.66	89.92	88.89
Ash	5.30	4.72	4.69
Crude protein	18.03	19.82	18.64
Fat	4.75	5.17	5.98
Crude fiber	4.30	3.54	3.15
Lactose	-	6.11	0.00
Starch	37.24	37.94	41.77
Ca	0.92	0.52	0.62
P	0.52	0.53	0.51
Na	NA	0.29	0.17
NE	2300	2519	2520
Lys	1.05	1.44	1.34
SID Lys	0.96	1.32	1.23

Abbreviation: NA: not available; SID: Standardized Ileal Digestible.

**Table 3 animals-12-02513-t003:** Primers for cytokines.

	Primer Forward 5′ → 3′	Primer Reverse 5′→ 3′	References
IFN-α	5′-CCCCTGTGCCTGGGAGAT-3′	5′-AGGTTTCTGGAGGAAGAGAAGGA-3′	Moue et al. [12]
IFN-γ	5′-TGGTAGCTCTGGGAAACTGAATG-3′	5′-GGCTTTGCGCTGGATCTG-3′	Royaee et al. [13]
TNF-α	5′-ACTCGGAACCTCATGGACAG-3′	5′-AGGGGTGAGTCAGTGTGACC-3′	Gabler et al. [14]
IL-12p35	5′-AGTTCCAGGCCATGAATGCA-3′	5′-TGGCACAGTCTCACTGTTGA-3′	Moue et al. [12]
IL-12p40	5′-TTTCAGACCCGACGAACTCT-3′	5′-CATTGGGGTACCAGTCCAAC-3′	Kim et al. [15]
IL-10	5′-TGAGAACAGCTGCATCCACTTC-3′	5′-TCTGGTCCTTCGTTTGAAAGAAA-3′	Royaee et al. [13]
TGF-β	5′-CACGTGGAGCTATACCAGAA-3′	5′-TCCGGTGACATCAAAGGACA-3′	Moue et al. [12]
IL-8	5′- GCTCTCTGTGAGGCTGCAGTTC-3′	5′-AAGGTGTGGAATGCGTATTTATGC-3′	Bracarense et al. [16]
IL-1α	5′- GTGCTCAAAACGAAGACGAACC-3′	5′-CATATTGCCATGCTTTTCCCAGAA-3′	Verpoest et al. [17]
IL-1β	5′-AACGTGCAGTCTATGGAGT-3′	5′-GAACACCACTTCTCTCTTCA-3′	Borca et al. [18]
IL-6	5′-CTGGCAGAAAACAACCTGAACC-3′	5′-TGATTCTCATCAAGCAGGTCTCC-3′	Borca et al. [18]

**Table 4 animals-12-02513-t004:** Lactation results including both batches and excluding gilts.

Parameters	CON	LSB	SEM	*p*-Value
Lactation days, #	25	25.2	-	-
ADFI, kg/d	5.38	5.32	0.031	0.312
Av IBW, kg	1.50	1.45	-	-
Weaned, #	11.3	11.4	0.203	0.971
Av FBW, kg	6.16	6.64	0.117	0.045
ADG, g/d	188	208	3.661	0.009

Abbreviations: #: number; ADFI: average daily feed intake; Av IBW: average initial body weight; Av FBW: average final body weight; ADG: average daily gain. Treatments: CON: control (commercial lactation diet); LSB: control + 12 × 10^9^ CFU/d of *S. cerevisiae boulardii* CNCM I-1079.

**Table 5 animals-12-02513-t005:** Performance results of sampled pigs by sow diet and piglet diet.

	Sow Diet			Piglet Diet			*p*-Value	
	CON	LSB	SEM	CON	LSB	SEM	Sow Diet	Piglet Diet
N, #	30	29	-	31	28	-	-	-
BW day 26, kg	14.42	14.46	-	14.52	14.35	-	-	-
BW day 49, kg	26.57	27.69	0.317	26.49	27.92	0.313	0.090	0.031
ADG days 26–49, g/d	527	576	13.79	524	586	13.62	0.090	0.031
BW day 70, kg	41.18	40.44	0.540	40.46	41.27	0.543	0.506	0.474
ADG days 50–70, g/d	745	676	21.99	702	723	22.61	0.128	0.655
ADG days 26–70, g/d	629	617	13.35	606	645	13.18	0.673	0.157

Abbreviations: #: number; BW: body weight; ADG: average daily gain. Treatments: Sow diet CON: commercial lactation diet; Sow diet LSB: control + 12 × 10^9^ CFU/d of *S. cerevisiae boulardii* CNCM I-1079; Piglet diet CON: control (commercial post-weaning diets); Piglet diet LSB: control + 2 × 10^9^ CFU/kg and 1 × 10^9^ CFU/kg feed in Prestarter and Starter, respectively, of *S. cerevisiae boulardii* CNCM I-1079.

**Table 6 animals-12-02513-t006:** IgA producing cells per sow treatment and tissue by sow diet and piglet diet (n/10000 µm^2^).

	Sow Diet			Piglet Diet			*p*-Value	
	CON	LSB	SEM	CON	LSB	SEM	Sow Diet	Piglet Diet
Intestine	25.76	22.73	1.60	23.29	25.38	1.60	0.460	0.785
Lung	3.97	3.69	0.36	3.90	3.77	0.36	0.370	0.807
Lymph node	6.28	4.14	0.68	5.85	4.61	0.68	0.039	0.085

Treatments: Sow diet CON: commercial lactation diet; Sow diet LSB: control + 12 × 10^9^ CFU/d of *S. cerevisiae boulardii* CNCM I-1079; Piglet diet CON: control (commercial post-weaning diets); Piglet diet LSB: control + 2 × 10^9^ CFU/kg and 1 × 10^9^ CFU/kg feed in Prestarter and Starter, respectively, of *S. cerevisiae boulardii* CNCM I-1079.

**Table 7 animals-12-02513-t007:** Cytokines gene expression results in lung, lymph node and intestine by sow treatment (fold).

	Lung				Lymph Node				Intestine			
	CON	LSB	SEM	*p*-Value	CON	LSB	SEM	*p*-Value	CON	LSB	SEM	*p*-Value
IFN-α	15.78	8.00	0.450	0.023	2.35	2.58	0.126	0.586	3.33	3.72	0.208	0.551
IFN-γ	20.10	10.42	0.522	0.037	4.79	4.67	0.131	0.570	5.33	5.65	0.151	0.257
IL-1α	12.85	3.90	0.459	0.029	1.23	1.66	0.088	0.074	1.97	2.03	0.101	0.821
IL-1β	10.78	2.76	0.463	0.024	2.24	2.87	0.160	0.303	3.81	4.09	0.196	0.618
IL-6	14.70	5.92	0.469	0.022	1.19	1.23	0.075	0.789	3.53	3.57	0.121	0.914
IL-8	10.77	1.85	0.446	0.041	6,36	6.84	0.211	0.604	5.40	5.75	0.134	0.148
IL-10	10.46	2.12	0.440	0.030	8.67	9.46	0.207	0.516	3.09	3.06	0.116	0.888
IL-12p35	12.17	3.62	0.439	0.045	1.90	1.78	0.099	0.554	15.04	15.32	0.113	0.334
IL-12p40	10.83	2.01	0.455	0.021	2.62	2.57	0.100	0.783	15.08	14.09	0.269	0.699
TNF-α	11.85	5.79	0.455	0.014	5.45	5.74	0.238	0.642	20.81	20.72	0.269	0.699
TGF-β	10.51	1.65	0.439	0.029	1.61	1.66	0.052	0.566	2.90	3.05	0.086	0.357

Treatments: Sow diet CON: commercial lactation diet; Sow diet LSB: control + 12 × 10^9^ CFU/d of *S. cerevisiae boulardii* CNCM I-1079; Piglet diet CON: control (commercial post-weaning diets); Piglet diet LSB: control + 2 × 10^9^ CFU/kg and 1 × 10^9^ CFU/kg feed in Prestarter and Starter, respectively, of *S. cerevisiae boulardii* CNCM I-1079.

## Data Availability

The datasets of the current study are available from the corresponding author on reasonable request.

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
