# Peer review of "Effect of Feeding *Saccharomyces cerevisiae boulardii* CNCM I-1079 to Sows and Piglets on Piglets’ Immune Response after Vaccination against *Actinobacillus pleuropneumoniae"

_animals, 2022, doi:10.3390/ani12192513_

Round 1

Reviewer 1 Report

In this work, the authors declare that feed lactating sows with the live yeast improves litter growth and weaning weight. The piglets from supplemented sows show higher mucosal protection and less inflammation in the lung after the vaccine. This paper is really well written, and the evidence presented in support of the authors' opinions is convincing. Here are some suggestions for reference.

1. Could IgA in the colostrum of sows, or in the intestinal mucus of sows and piglets been detected quantitatively?

2. Were the infection of pathogens about the sows clear? Will this affect the results?

3. Could the cytokines in lung, lymph node and intestine been detected quantitatively

Author Response

Dear Reviewer.

Thank you for the time spent reading our paper, and for your questions/suggestions, which we are replying hereby:

  1. Could IgA in the colostrum of sows, or in the intestinal mucus of sows and piglets been detected quantitatively? Unfortunately, we have not quantitatively measured IgA in colostrum or intestinal mucosa. However, one of the references of the paper reports that feeding the same live yeast as we did to lactating sows, tended to increase Immunoglobulin A in colostrum. Regarding intestinal mucus, our assumption was that the number of IgA producing cells is correlated to the IgA concentration. Therefore, since the technique to detect the number of IgA producing cells is well described and validated by the Research Team of the University of Murcia, we decided not to measure IgA concentration.
  2. Were the infection of pathogens about the sows clear? Will this affect the results? Before the trial, we checked in the farm the presence of any wild strain of Actinobacillus pleuropneumoniae (APP), with 2 goals: i) to ensure that the first contact of the animals with APP was the vaccination, and ii) to avoid potential bias in the results in case we decided to measure specific antibodies against APP (in case of the presence of a wild strain we could have faced a confounding situation). Finally we did not measure specific IgG due to supplier recommendation, but how vaccination response is affected with live yeast supplementation is definitely something to look at in the near future.
  3. Could the cytokines in lung, lymph node and intestine been detected quantitatively? They can be quantitatively detected by ELISA. But we did not. The mRNA expression that we measured is really a "photography" of what is happening in that moment, and that is why we went for that approach. A posteriori, we have discussed about the added value of having measured the inflammatory status in different timepoints along the experiment to have a longitudinal study, but unfortunately we could only look at the timepoint 3 weeks after the vaccination pattern was complete (3 weeks after the respiratory challenge).

Hoping these explanations and answers satisfy your remaining interest.

Sincerely,

The Authors 

Reviewer 2 Report

Refer annotated manuscript for Reviewer comments.

Author Response

Dear Reviewer.

Thanks for the time to read and evaluate the manuscript, and for your valuable comments. We have upload a second version of the manuscript with all the editions you proposed. The function "Track Changes" in MS Word has been used, therefore all the modifications are visible in the new version. Additionally, please find below our answers and comments to your questions/suggestions:

  •  Line 45: Is high-quality pigs referencing growth performance? We have references about the effects of sow nutrition on growth performance. However, as indicated at the end of the paragraph, high-quality may involve piglets better prepared in terms of immune status. All together improves the efficiency of the production system.
  • Lines 70-72: we have reworded the statement to make it less strict. Obviously we have reviewed lots of publications and no evidence of similar works, but the word "scarce" seems to be more in the safe side.
  • Line 95: we have modified the probiotic dose according to what is more commonly reported.
  • Lines 97-98: we have included the mean parity number for supplemented and non-supplemented sows. 
  • Lines 103-105: BW and BFT were measured with the objective of allocation into treatments, and it is mentioned before in Line 92. Therefore it is removed and the device to measure BFT is indicated in Line 92.
  • Materials & Methods: this section has been improved for clarity.
  • Lines 180-181: sentence removed because it is indicated later in the results.
  • Lines 243-244: it is true that the duration of supplementation. Indeed, in research facilities or commercial facilities with a different management, the goal is to supplement from even earlier than the current study. We did not find external peer review publications with significant results in weaning weight for a similar supplementation period. We have reworded the sentence accordingly. 
  • Lines 261-262: actually the CON piglets grew only numerically faster than the LSB-piglets. Therefore, we cannot really say that they out grew. However, the change of facilities is always a challenge. Under good hygiene conditions as in the post-weaning facility, it could be that the LSB piglets grew more due to the lower E expenses in generating immune compounds and fighting against hostile environment. If that was the case, the change of facilities may have had negative consequences for them, since it is a possibility that they are less immune-stimulated to face the new situation. The dissertation in the following paragraph is about that. 
  • Lines 275-289: we have rewritten to clarify. It is hard to say that the sample size was not enough to depict differences, since in the immediately previous period the same sample size allowed us to obtain significant differences. However, it is true that looking at the SEM´s in Table 5, the varibility in the population seems bigger after the second vaccination than before. Thus, it could be that for ADG between days 50 and 70 the power is limiting, and partially explaining the lack of difference.
  • Line 307: in the farm where the trial took place they do not vaccinate against APP. Prior to the start of the trial, a group of sows was randomly blood sampled and checked for the presence of circulating antibodies and APP. Nothing was found.
  • Lines 363-368: It is definitely a speculation we recognize that in the text (Lines 363-364: "In our study, although there were no microbiota measurements..."). However, The Authors believe that one of the key facts explaining the results is the gut microbiota modulation by the live yeast and the gut-lung axis: i) we have indicated several references about the connection between gut microbiota and respiratory issues (35, 36, 37, 38, 39); ii) we have internal data about the effect of live yeast supplementation to piglets on their gut microbiota, and published (30) data about the effect of the live yeast supplementation to sows on piglets´ microbiota, and iii) we very much support the hypothesis described in the manuscript. The limitation is that we have not study in depth gut microbiota, but that is why we have discussed the results this way. Next step we are working on is in investigating all together in the same study; it is a topic that really deserves time.
  • Lines 406-407: The sentence has been removed. We consider it can be a repetition of the previous lines that are postulating the hypothesis.

Hoping that the answers satisfy your comments,

Sincerely,

The Authors

Round 2

Reviewer 2 Report

I am satisfied that the authors have considered the reviewer comments.